# Sensitivity of the global carbonate weathering carbon-sink flux to climate and land-use changes

Sibo Zeng [1], Zaihua Liu [2,3]* & Georg Kaufmann [1]*

The response of carbonate weathering carbon-sink flux (CCSF) to its environmental drivers is still not well understood on the global scale. This hinders understanding of the terrestrial carbon cycle. Here, we show that there is likely to be a widespread and consistent increase in the global CCSF (ranging from $+9.8\%$ (RCP4.5) to $+17.1\%$ (RCP8.5)) over the period 1950–2100. In the coming years the increasing temperature might be expected to have a negative impact on carbonate weathering. However, the increasing rainfall and anticipated land-use changes will counteract this, leading to a greater CCSF. This finding has been obtained by using long-term historical (1950–2005) and modeled future (2006–2100) data for two scenarios (RCP4.5 and RCP8.5) for climate and land-use change in our CCSF equilibrium model. This study stresses the potential role that carbonate weathering may play in the evolution of the global carbon cycle over this century.

[1] Institute of Geological Sciences, Geophysics Section, Freie Universität Berlin, 12249 Berlin, Germany. [2] State Key Laboratory of Environmental Geochemistry, Institute of Geochemistry, CAS, 550081 Guiyang, China. [3] CAS Center for Excellence in Quaternary Science and Global Change, 710061 Xi'an, China. *email: liuzaihua@vip.gyig.ac.cn; georg.kaufmann@fu-berlin.de

There are huge uncertainties in the response of the terrestrial carbon cycle to changing environmental conditions, such as global warming and human intervention[1,2]. A growing body of evidence indicates that contemporary continental weathering processes are sensitively responding to climate change and human activities[3–6]. The carbonate weathering carbon sink, about 0.2–0.7 Gt C yr$^{-1}$, is an important component of the global carbon budget, accounting for ~7–25% of the estimated terrestrial carbon sink[5,7,8]. The rapid kinetics driving carbonate weathering (reaching equilibrium in three hours under experimental conditions[9]) results in dissolution rates nearly 15 times faster than those of silicate rocks[10], thereby responding quickly to environmental fluctuations. The chemical weathering of carbonate rocks is a complex terrestrial process that is controlled by numerous natural and anthropogenic drivers[5–9]. To summarize and simplify the mixed impacts of all drivers, a generic equation for the carbon-sink flux produced by carbonate weathering can be expressed as[11]:

$$CCSF = 0.5 \times 12 \times R \times DIC \qquad (1)$$

where CCSF (t C km$^{-2}$ yr$^{-1}$) is the carbonate weathering carbon-sink flux, $R$ is runoff in m yr$^{-1}$, and DIC (mmol L$^{-1}$) is the concentration of dissolved inorganic carbon produced by carbonate weathering; 12 is the molar atomic weight of carbon, and the ratio 0.5 indicates that only one half of the $HCO_3^-$ generated by carbonate weathering is of atmospheric origin[5].

Previous work has highlighted the diverse geochemical, climatic and ecological factors that influence both $R$ and DIC, and thus the CCSF variations, including (amongst others) surface temperature[6], precipitation and runoff[11,12], net primary production of ecosystem and soil $CO_2$[6,13], carbonate lithologies[12], atmospheric $CO_2$ concentration[14], soil water content[15] and land-use patterns, and practices[4,16,17]. In natural environments, these factors are tightly interwoven and controlled by climate and land cover[17,18]. Recently, studies on different spatial scales have reported that climate perturbations and human interventions have dramatically changed CCSF over the past few decades. For example, in the Mississippi River basin the increased rainfall, high proportion of cultivated area, water management, and use of lime for fertilization have remarkably enhanced the $HCO_3^-$ export flux, with a nearly +50% increase in the recent decades[4]. In addition, the N-fertilizer uses for agriculture also produced nitric acid which enhanced $HCO_3^-$ flux as a $CO_2$ source[19]. The $HCO_3^-$ and ground water $CO_2$ storage of a karst aquifer in Konza Prairie (central USA) displayed synchronous increases during the past 26.5 years, which was attributed to the long-term changes of temperature and land use[16]. In northern high latitudes, two large Russian arctic rivers have experienced major increases (135–180%) in alkalinity due to climate change and anthropogenic impacts during the past 40 years[20]. However, there are also reports of a decline of CCSF in some other regions. For instance, in the typical monsoon region of Southwest China, a model study found that climate change (especially, reduced rainfall) caused a 19% decrease in CCSF during the past 40 years[12]. In sum, although these individual studies have detected regional CCSF perturbations attributed to one or a few environmental drivers, a comprehensive analysis of the overall global CCSF fluctuation in response to all driving factors is still lacking. In particular, to our knowledge no studies have considered the impacts of long-term land-use change on CCSF fluctuations at the global scale. In many areas with intensive human intervention, land-use changes have altered the CCSF by changing the runoff patterns and affecting the soil $pCO_2$ through changing, amongst others, the productivity and soil properties[17], etc.

Here, we explore the spatiotemporal CCSF variations on global carbonate rock outcrops by constructing a mixed-effect model that considers the interrelated impacts of climate and land-use dynamics. We provide a comprehensive interpretation of environmental impacts on CCSF fluctuations by analyzing the spatialtemporal relationship over a lengthy historical period, 1950–2005. We further predict the response of CCSF to the changes in temperature, precipitation, and land use that are presented in the Coupled Model Intercomparison Project Phase 5 (CMIP5) climate projection, adopting two of its representative scenarios, RCP4.5 and RCP8.5. CMIP5 is trying to predict future climate by estimating the amounts of atmospheric carbon dioxide that will be produced in the future. Different RCPs predicting the radiative forcing achieved by the year 2100 AD range from 2.6 to 8.5 (RCP2.6–2RCP8.5) watts per square meter (Wm$^{-2}$). Here, RCP4.5 is selected to be representative of the moderate-stabilized emission scenarios (medium $CO_2$ increase), whereas RCP8.5 represents the more aggressive scenarios (large $CO_2$ increases). Based on these choices, we attempt to predict the coupled effects of current major shifts in climate and land use on the CCSF fluctuations in the future. We reveal the sensitivity of the CCSF response to the above-mentioned drivers in different latitudinal regions and estimate the role played by carbonate rock weathering in the global carbon cycle over the remainder of this century.

## Results

**General overview.** In this section, we show the results of our CCSF model first at a global scale and then focus on drivers that will vary at broad regional scales. As a first step, soil $CO_2$ pressure ($pCO_{2(soil)}$) is derived from (3) and ET (evapotranspiration) is based on Eq. (6) (see Methods). Accordingly, we can calculate the calcium equilibrium concentration $[Ca^{2+}]_{eq}$ from Eq. (2) (see Methods), and the $R$ from the difference between precipitation (P) and ET. Next, we extract the $[HCO_3^-]$ and P-ET for each grid cell located on a carbonate outcrop to obtain the CCSF by Eqs. (1) or (7) (see Methods), then sum to obtain the total carbon sink (TCS) budget using Eq. (8) (see Methods). We consider that these results can help us to estimate the feedback of CCSF response to climate and land-use change under the different future scenarios envisioned by CMIP5, thereby evaluating the role that carbonate weathering will play in the global carbon cycle in the future.

**Overall fluctuation in $[HCO_3^-]_{eq}$, $R$, and CCSF.** In Fig. 1a, b, we present the overall changes of the two fundamental CCSF drivers, $[HCO_3^-]_{eq}$, and runoff ($R$), over the full model period. $[HCO_3^-]_{eq}$ displays steadily increasing trends of +2.1–+2.6% from 1950 to 2100. The larger $[HCO_3^-]_{eq}$ increase is found in scenario RCP8.5, with about +0.0006 mmol L$^{-1}$ yr$^{-1}$ (Fig. 1a). The amplitude of global runoff variations, by contrast, is 5.7–8.0 times larger than the $[HCO_3^-]_{eq}$ in the same period (Fig. 1b), with runoff increasing at around +0.18 mm yr$^{-2}$ for the historical period, a finding that is close to other published results[21,22]. For the full period, runoff from carbonate rocks increases around +12.0% (RCP4.5) or +20.9% (RCP8.5). After summing these two drivers by using Eq. (8) (see Methods), we found a widespread and consistent increase in global CCSF, with values around +9.8% (RCP4.5) or +17.1% (RCP8.5) at the end of this century (Fig. 1c). As with runoff, the CCSF increase under RCP8.5 (0.0068 t C km$^{-2}$ yr$^{-2}$) is higher than under RCP4.5 (0.0043 t C km$^{-2}$ yr$^{-2}$).

**Spatial differences in CCSF and its long-term trend.** To determine which areas have experienced significant CCSF changes, particularly the areas that are mainly responsible for the calculated increases, we now consider the different geographical regions. Figure 2a summarizes the spatial annual mean CCSF at the global scale for the historic period. Mean annual CCSF ranges from 0.06 t C km$^{-2}$ yr$^{-1}$ in the arctic regions to

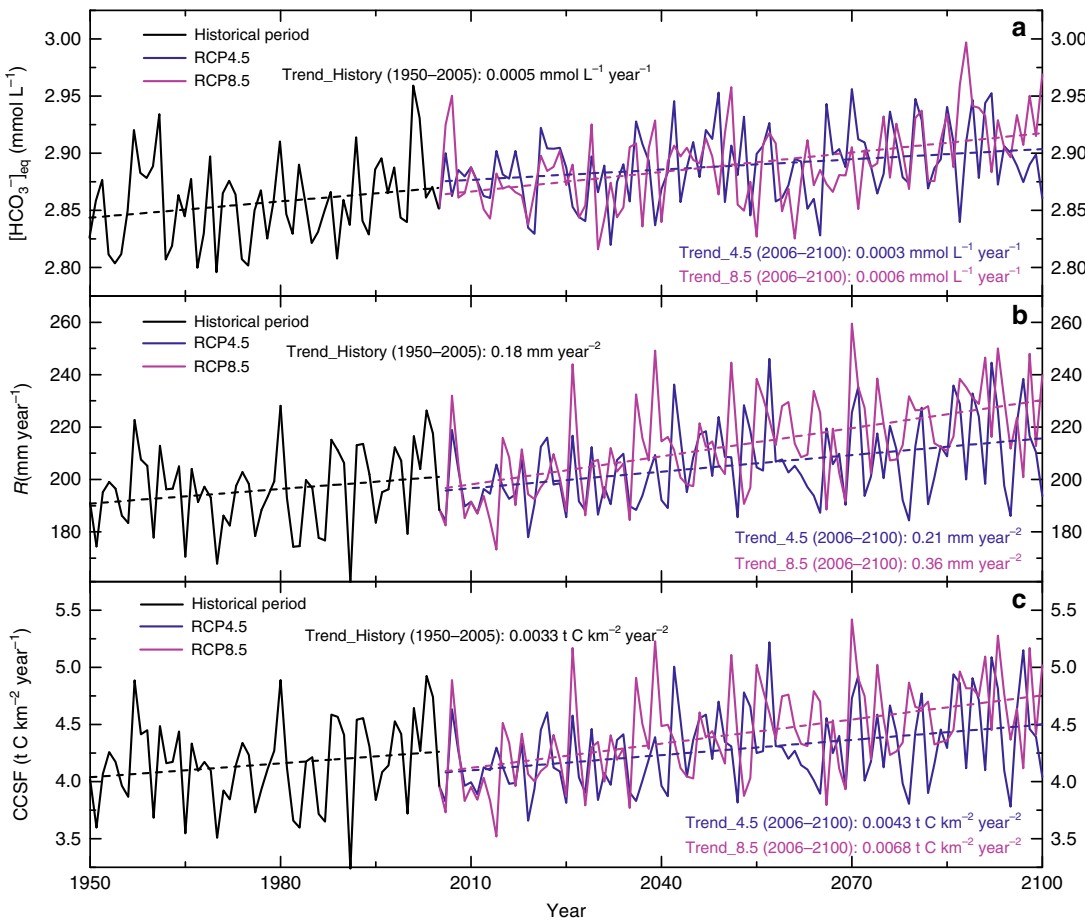

**Fig. 1 Interannual changes in relevant variables. a** $[HCO_3^-]_{eq}$, **b** $R$ (runoff), and **c** CCSF (carbonate weathering carbon-sink flux) on global carbonate rock outcrops during the historical period (1950–2005) and the two future (2006–2100) scenarios (RCP4.5 and RCP8.5). All the variables display increasing trends. The historical period (black line) has the lowest CCSF variation and RCP8.5 (purple line) has the highest, indicating the substantial response of CCSF to dramatic climate change and land-use conversion.

46.42 t C $km^{-2}$ $yr^{-1}$ in and near the equatorial regions. We observe prominent spatial differences, with the highest CCSF occurring in tropical areas and temperate to subtropical humid areas, such as Southwest China, North America, and West Europe, whereas the lowest CCSF occurs mostly in the arctic regions and arid areas, e.g., Central Asia and Saharan Africa. We use spatial linear regression analysis to extend the spatialtemporal CCSF trends of 1950–2005 to 2006–2100. The two RCP scenarios show similar spatial CCSF trends. The strongest CCSF increases occur in most tropical regions and also in North America, West Europe, and Tibet (Fig. 2b, c). The CCSF under RCP8.5 displays similar but stronger increases in most of the areas than does RCP4.5. There are negative effects in the Middle East and North Africa, as these regions experience CCSF decrease due to a drier climate.

**Latitudinal change of CCSF, $R$, and $[HCO_3^-]_{eq}$ trends**. Next, we consider spatial CCSF changes by summarizing the latitudinal variation trends of soil $pCO_2$, CCSF, $R$, and $[HCO_3^-]_{eq}$ (Fig. 3). This approach can help us to get a better understanding of how the regional CCSF responses to climate and land-use change may differ during the two periods (historical, and future under RCP4.5 and RCP8.5). As shown in Fig. 3a, the soil $pCO_2$ increasing trends in high latitudes are generally higher than those in the low latitudes. RCP8.5 scenarios show a larger $pCO_2$ increase. Figure 3b demonstrates the modelled $[HCO_3^-]_{eq}$, which shows consistent increasing trends in cool and humid regions, such as the mid and high

latitudes, but decreasing trends in lower latitudes. The more dramatic climate and land-use change scenario of the future (RCP8.5) results in a stronger negative $[HCO_3^-]_{eq}$ trend in low latitudes, and a more positive trend in high latitudes ($-0.0005$ mmol $L^{-1}$ $yr^{-1}$ versus 0.0025 mmol $L^{-1}$ $yr^{-1}$). In contrast, runoff shows rising trends generally, especially at low latitudes under the two RCP scenarios (Fig. 3c), where there is a high proportion of land-use change from forest to crop. The latitudinal CCSF variation as shown in Fig. 3d behaves like the runoff changes, showing an increase in low latitudes and being 2.85–6.25 times greater than in high latitudes. Although $[HCO_3^-]_{eq}$ concentrations in high latitudes will experience dramatic increases, the CCSF variations in these regions are less significant when compared to their values in low latitudes. The southern mid latitudes are interesting regions, as here the changes are considerable. However, due to the small proportion of carbonate rock outcrops there (1.6%), those changes are less important for the global carbon-sink budget.

## Discussion
From the Results section above, we have found that the coupling between natural and anthropogenic factors in different latitudinal zones results in large differences in the regional CCSF response. Thus, a better understanding of the sensitivity of carbonate weathering carbon flux to its different environmental drivers is crucial for estimating the role of CCSF in the global carbon cycle in the future. Therefore, the causes of CCSF variations under the climate and land-use change in different areas will be explored next.

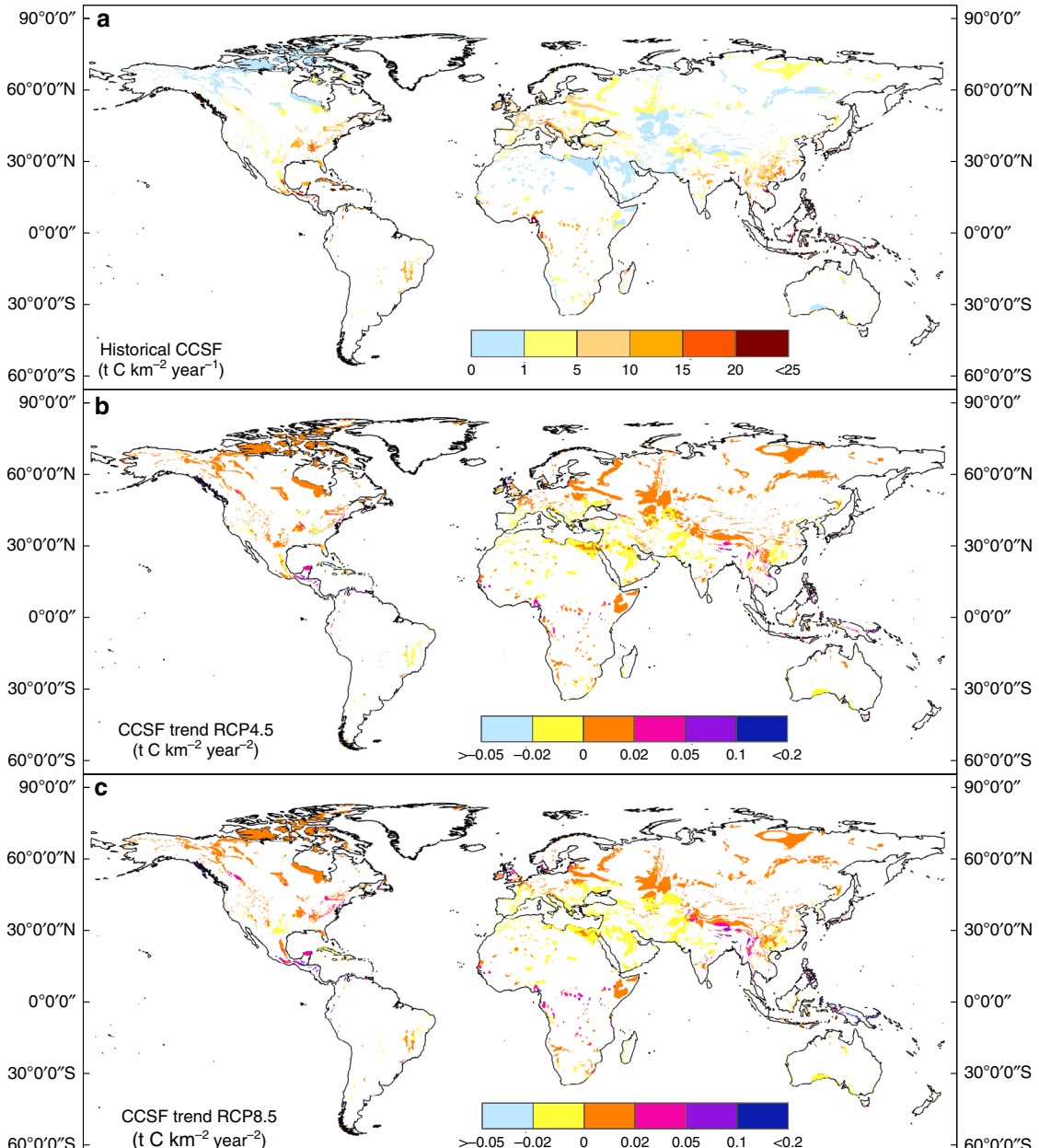

**Fig. 2 Spatial distribution of CCSF and its changes. a** Annual average CCSF (carbonate weathering carbon-sink flux) in carbonate rock outcrops for the historical period (1950–2005) and its changes for the two differing climate and land-use change scenarios, **b** RCP4.5, and **c** RCP8.5. Note: nearly 72% of carbonate rock outcrop is distributed in the mid and high latitudes (30°–90°) and less in the low latitudes (0°–30°).

First, we made a comparison of modelled CCSFs with observed global data. The aim here is to test the accuracy of our model estimates of CCSF changes in the different climatic and land-use patterns around the world. Table 1 compares our results to other studies to check reliability. Our modelled CCSF variations are in good agreement with a variety of independent carbonate weathering carbon-flux estimates around the world, including those from the full range of latitudinal zones and with distinct climate and land-use conditions: the difference (error) is generally <10%. Accordingly, we judge that our model reliably predicts spatial CCSF differences and can be used for future estimation.

If our global mean CCSF (4.3 t C km$^{-2}$ yr$^{-1}$) is applied to the global carbonate area (i.e., ~50% of the continent surface[5]), we obtain a total annual global carbon sink of 0.32 Gt C yr$^{-1}$.

Temperature is a fundamental controlling factor in carbonate weathering as demonstrated by many studies[6,9,15]. Generally, it is

found that [HCO$_3^-$] variation is highly sensitivity to temperature, reaching maximum values in the temperature range (10–15 °C), i.e., both very low and high temperatures will limit carbonate weathering[6,15]. This behavior is a result of competition between thermodynamic control of the weathering and the variability of soil CO$_2$ production by soil biota[6,15]. [HCO$_3^-$]$_{eq}$ will be positively correlated to temperature below 15 °C (Fig. 4a). In the inter-tropical zone, the warm temperatures may considerably decrease the [HCO$_3^-$]$_{eq}$. This is confirmed by inspecting the latitudinal trends of [HCO$_3^-$]$_{eq}$. For instance, the strongest warming trends (+0.015 °C yr$^{-1}$ to +0.023 °C yr$^{-1}$) in high latitudes will significantly increase the [HCO$_3^-$]$_{eq}$ there. In contrast, rising temperatures in low latitudes will limit the carbonate dissolution, which results in a negative [HCO$_3^-$]$_{eq}$ trend (Fig. 3b). However, according to our results, latitudinal [HCO$_3^-$]$_{eq}$ variations do not always follow temperature variations alone. The impacts of

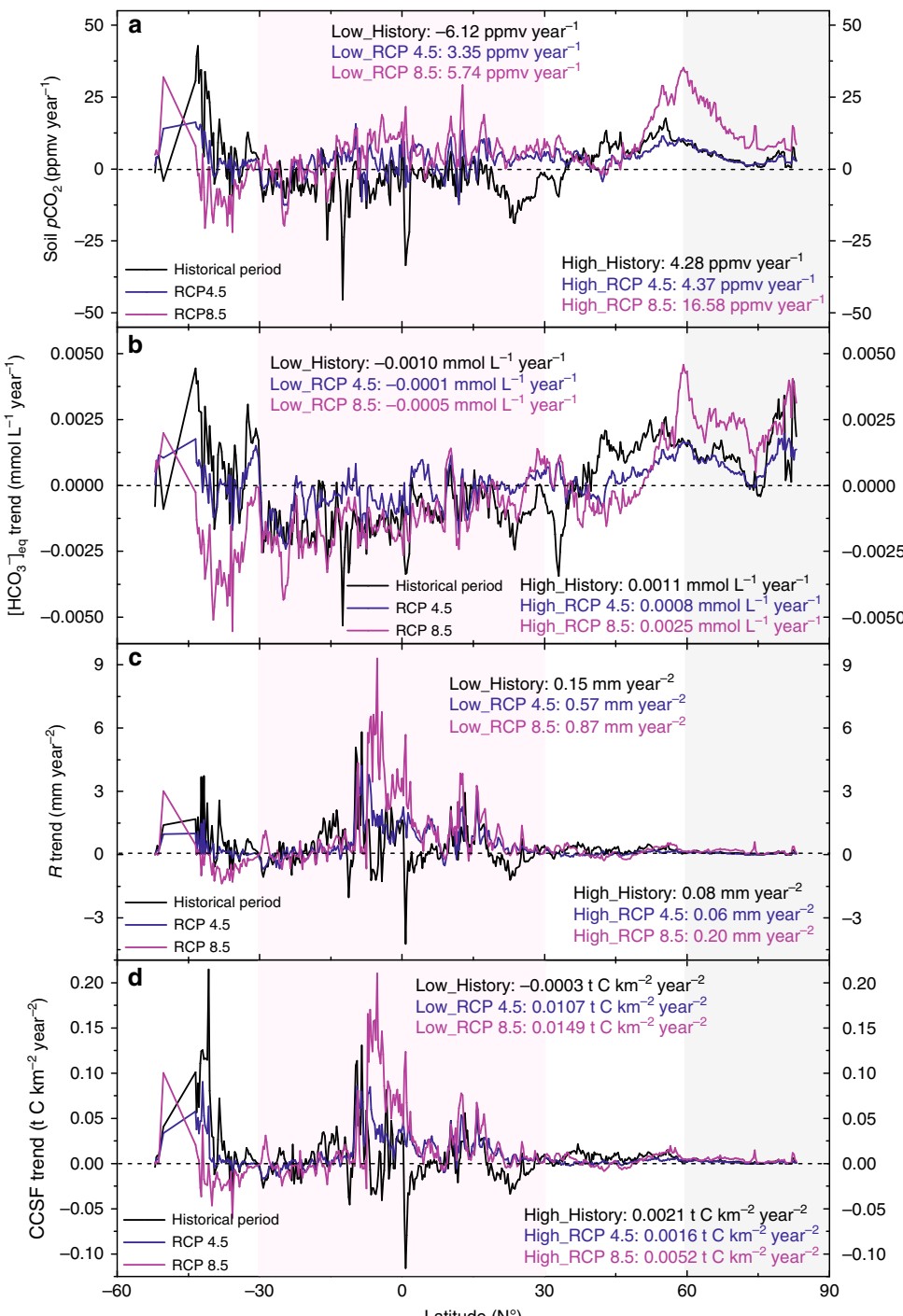

**Fig. 3 Latitudinal distribution of relevant variable trends. a** Soil $pCO_2$, **b** $[HCO_3^-]_{eq}$, **c** $R$ (runoff), and **d** CCSF trends for three cases (the historical period, and the future period for RCP4.5 and RCP8.5). The shaded areas are the northern high latitudes (60°N–90°N, light gray) and the low latitudes (30°S–30°N, light pink).

changing precipitation and land use control soil $pCO_2$ distribution (Fig. 4d). Discussed together with temperature, these factors are also equally significant and therefore control the actual global $[HCO_3^-]_{eq}$ distribution (Fig. 4b, c). For example, we observe three $[HCO_3^-]_{eq}$ peaks on the global graph (Fig. 4a). Two of them are not located in the theoretical region of maximum dissolution (10–15 °C), a feature that has not received much attention. We argue that the higher $[HCO_3^-]_{eq}$ in these regions is mainly caused by changes in land-use patterns (Fig. 4c), soil $pCO_2$ (Fig. 4d), and increased precipitation (Fig. 4b).

According to our analysis, the CCSF fluctuations are strongly depending on the runoff, rather than on $[HCO_3^-]_{eq}$ or temperature (Figs. 1–3) alone. Precipitation, temperature, and vegetation cover are key factors that control runoff in many models (Fig. 4) and thus also CCSF variations.

We employ long-term spatial regression analysis to detect relationships between CCSF and the variables, runoff, and equilibrium $HCO_3^-$ concentration. Figure 5 compares the individual impacts of $[HCO_3^-]_{eq}$, and $R$ on the annual CCSF fluctuations. The results show that the regional variations of CCSF were

**Table 1 Comparison of CCSF between our modelled results and other studies in different latitudinal zones with different climate and land-use conditions.**

| Location | Latitudinal zone (low/mid/high) | $T$ (°C) | $P$ (mm yr$^{-1}$) | Main land-use type | CCSF (t C km$^{-2}$ yr$^{-1}$) in other study | This study |
|---|---|---|---|---|---|---|
| Guizhou | Low | 15 | 1225–1425 | Forest/crop/grass | 7.86–10.90[(1)] | 7.63–11.16 |
| Xijiang | Low | 14–22 | 800–1200 | Forest/crop/grass | 7.31[(2)] | 7.30 |
| Kikori | Low | 21 | 4330 | Forest | 29.36[(3)] | 29.19 |
| Thailand | Low | 26 | 3168 | Forest | 42.30[(4)] | 40.60 |
| Puerto Rico | Low | 24 | 2100 | Forest/grass/crop | 19.77[(5)] | 28.24 |
| Florida | Low | 21.1 | 1336 | Forest | 9.49–10.05[(6)] | 11.04 |
| Slovenia | Mid | 6–11 | 800–3000 | Forest/grass/crop | 15.16–32.89[(7)] | 16.49–26.76 |
| Southern Alps | Mid | 9 | 1300 | Forest | 11.91[(8)] | 11.58 |
| Siberia | High | −7 to −14 | 250–400 | Forest/non forest | 1.52–2.15[(9)] | 1.49–3.30 |
| Mackenzie | High | −1 | 250–1500 | Forest/non forest | 4.94[(10)] | 3.44 |

References: (1) Zeng et al.[11]; (2) Xu and Liu[38]; (3) Ferguson et al.[39]; (4) Pitman[40]; (5) Giusti[41]; (6) Moore et al.[42]; (7) Szramek et al.[43]; (8) Sarazin & Ciabrini[44]; (9) Huh et al.[45]; and (10) Millot et al.[46].
Note: the higher CCSF cited for the Mackenzie River basin in northern Canada may be due to sulfide oxidative weathering[46] contributing to the carbonate weathering, which does not contribute to the carbon sink but is possibly a $CO_2$ source

typically driven by trends in runoff (global mean $R^2 > 0.95$, $P < 0.001$) but not $[HCO_3^-]_{eq}$. The substantial variability of CCSF is responding to differing runoff, as noted also in other studies[11,12]. The reason why CCSF is more sensitive to runoff than to $[HCO_3^-]$ has been attributed chiefly to the chemostatic behavior of the latter[11].

To better explain the dominant control of this behavior, we divide global CCSF variations into three latitudinal zones (0°–30°, 30°–60°, and 60°–90°) with different mean temperatures, as shown in Fig. 6. $[HCO_3^-]$ shows a significant positive relationship with CCSF only for the high latitudes (60°–90°), while the correlation declines towards the equator (Fig. 6a). Runoff, however, shows a significant ($R^2 > 0.96$, $P < 0.001$) positive relationship with CCSF across all latitudinal zones (Fig. 6b). More importantly, it is noticed that when the $[HCO_3^-]_{eq}$ decreases in low latitudes due to global warming, the accompanying increase in runoff overwhelms the temperature effect, leading to net increases in CCSF. Therefore, based on the results of our model, we suggest that global CCSF variations are highly dynamic and mainly determined by the hydrological cycle (runoff).

For a long time human activities were not considered in global carbonate weathering models. However, recent studies[4,11,17] have found that land use does play a significant role in CCSF control and should be considered in carbon-sink models. On the one hand, consideration of land use can help us to explain why similar climate conditions present highly scattered $[HCO_3^-]_{eq}$ distributions in different datasets[6,15]. As indicated in Fig. 4, the latitudinal $[HCO_3^-]_{eq}$ curves should show similarities to temperature and/or precipitation trends if climatic factors are considered alone. However, we find that the three $[HCO_3^-]_{eq}$ peaks occur in three latitudinal zones (50–70°N, 0–10°S, and 40–50 S°) that have a high proportion of forest cover. Globally, as the proportion of forested areas increase, soil $pCO_2$ and $[HCO_3^-]_{eq}$ increases. In contrast, when grass and crop cover increase, soil $pCO_2$ and $[HCO_3^-]_{eq}$ decreases (Fig. 4c, d). Land-use change can also dramatically alter water balances. In northern high latitudes where precipitation is low and forest cover is high, runoff ($R$) decreases sharply (Fig. 4e–g). In contrast, the increasing cropland area in low latitudes drastically increases net runoff. Based on our simulation, the role of land-use change will be even more important in the future. From 2006 to 2100, cropland proportion in low latitudes will increase by a factor of two (from 8% to 16%), resulting in decreased $[HCO_3^-]_{eq}$ and increased runoff. In the historical period (1950–2005), mid and high latitudes dominated the increase of the annual TCS (100%, $7.7 \times 10^4$ t C yr$^{-1}$). During the continuing

climate and land-use changes expected in the future (2006–2100), this situation will reverse. Although the carbonate rock outcrops in low latitudes constitute only 28% of the terrestrial carbonate area, the higher sensitivity of CCSF to climatic and anthropogenic changes in these areas in the future will contribute 61-68% of the TCS increase ($5.6 \times 10^4$ t C yr$^{-1}$ to $8.1 \times 10^4$ t C yr$^{-1}$). More importantly, the drastic land-use transition (mainly to agricultural land use following deforestation) will contribute 42–50% of total TCS increase in spite of the $[HCO_3^-]_{eq}$ decline. Therefore, we stress that the CCSF shows great sensitivity to anthropogenic impacts. Human land-use activities will significantly alter the CCSF and are as important as climatic drivers in certain areas.

The global annual average temperature in carbonate regions in the historical period (1950–2005) was 17 °C, which already exceeded the temperature range for maximum carbonate dissolution (Fig. 4a). If global warming continues in the future, the higher global mean temperatures will constrain carbonate weathering. In low latitudes, although the climate change will promote the soil $CO_2$, land-use transitions to agriculture after deforestation in this warming background will decrease $[HCO_3^-]_{eq}$ in the future. Carbonate weathering will show less sensitivity to the overheated environment in these regions. Our results find that increasing precipitation will offset the negative impacts of temperature and deforestation there. In the future, we believe that the CCSF fluctuations will become larger, sensitively responding to climate and land-use changes, and the increasing carbonate weathering flux from terrestrial waters to oceans may promote the biological carbon consumption by organisms in these systems[5,23]. Therefore, this increasing flux can be a considerable carbon sink that against the rising atmospheric $CO_2$ concentration in the future, potentially become a negative feedback to global warming.

Our model still needs some improvements for future studies. For example, a growing body of evidence finds that the elevated $CO_2$ in the atmosphere ($CO_{2atm}$) will affect the primary productivity of ecosystems by the so-called $CO_2$ fertilization effect[24]. Rising $CO_{2atm}$ will also alter the soil $CO_2$ and water balances, and thus impact CCSF: it must be considered in global carbon-sink modeling. In addition, land-use change can prompt changes in subsurface flow paths and mineral water interaction, thus the resultant fluxes of solutes from landscapes. Given thermodynamic controls on carbonate weathering, water fluxes through the landscape will have the biggest control on $[HCO_3^-]_{eq}$, which need to be accounted for in land-use change dynamics. Meanwhile, anthropogenic N and S inputs from use of fertilizers[19] or

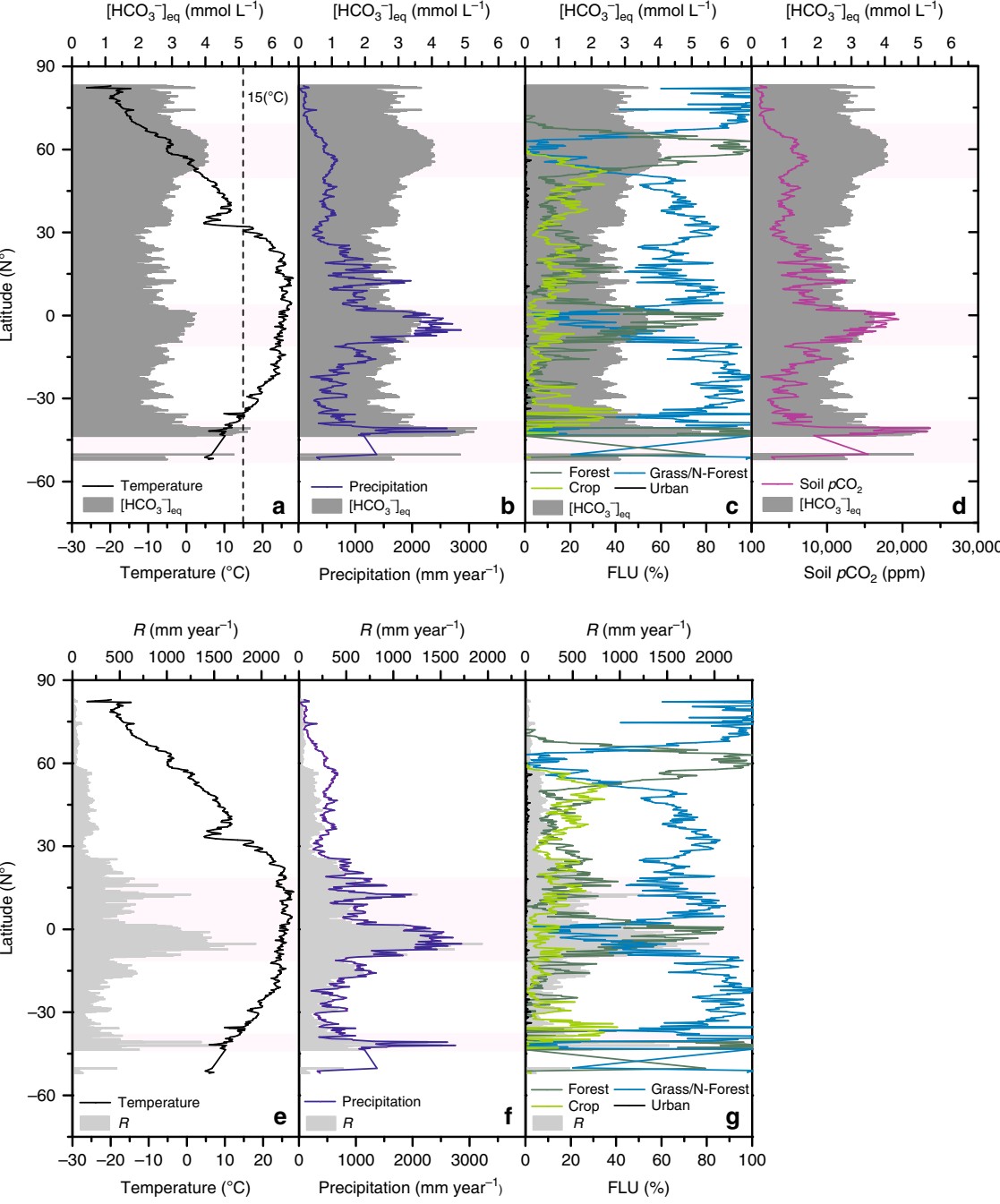

**Fig. 4 Latitudinal variations of relevant variables.** $[HCO_3^-]_{eq}$ (dark-shaded area in **a–d**) and $R$ (runoff, light-shaded area in **e–g**) in relation with mean temperature (black line in **a** and **e**), mean precipitation (blue line in **b** and **f**), land-use type (multicolor lines in **c** and **g**), and soil $pCO_2$ (purple lines in **d**) in the historical period (1950–2005). Three $[HCO_3^-]_{eq}$ peaks occur in three latitudinal zones (50–70 °N, 0–10 °S, and 40–50 S°). The dashed line in **a** is the upper temperature limit (15 °C) for maximum carbonate dissolution. The highest runoff ($R$) can be found in the tropical zone and the area close to 40 °S.

coal combustion[25] have become additional drivers of carbonate weathering. The carbonate dissolution produced by nitrate or sulfuric acids will lead to increased $[HCO_3^-]_{eq}$ as a $CO_2$ source. For example, Perrin et al.[19] found this $CO_2$ source by nitric acid due to agriculture contribution is not negligible, since it could reach 6–15% of $CO_2$ uptake by natural silicate weathering and could consequently partly counterbalance this natural $CO_2$ sink. However, to give an estimate of this flux in the future may be difficult, which is out of the focus of this contribution.

In this study, we have assembled a new model to explore spatialtemporal global CCSF fluctuations over the historical

period, 1950–2005, and extended it to the end of this century 2100 AD. Besides natural fluctuations in temperature and carbondioxide concentration, anthropogenic land-use changes have been considered. The results show that there will be widespread and consistent increases in global CCSF, ranging from +9.8% (RCP4.5) to +17.1% (RCP8.5), that are chiefly due to increasing runoff (+12% to +20.9%) and $[HCO_3^-]_{eq}$ (+2.1% to +2.6%). For the full period, 1950–2100, due to the increased runoff caused by both land-use transition and increasing rainfall, CCSF variations in low latitudes are expected to become the largest. Although the low latitudes contain only 28% of terrestrial

carbonate rock outcrops, the CCSF increase here accounts for 61–68% of the TCS in the future. The warming trend in mid and high latitudes will accelerate the carbonate dissolution but the total impact is less important. In future, the increase of runoff will dominate CCSF increases, due to the chemostatic behavior of $HCO_3^-$. Global warming, by contrast, will lead to lower $[HCO_3^-]_{eq}$ in tropical regions due to the warmer temperatures. However, land-use changes and the accompanying rise in water

flux could well counteract this impact, leading a higher net CCSF. Our study highlights the significant role of land-use change in global CCSF variation, which needs to be considered in future global CCSF models.

## Methods

**Selection of database.** To simulate the CCSF fluctuations from the historical period to the end of this century, we use a long-term statistical climate dataset from the NASA Earth Exchange Global Daily Downscaled Projections (NEX-GDDP) CMIP5 archive (Coupled Model Intercomparison Project Phase 5). This estimates spatiotemporal variations in climate change[26], including a global dataset of reconstructed (1950–2005) historical precipitation, maximum and minimum near-surface temperatures, and future predictions (along the concentration pathways, RCP4.5 and RCP8.5, from 2006 to 2100). We calculate the mean temperature by using the average value of daily maximum and minimum temperatures. From the NEX-GDDP model suite, we select the Earth System Model of the Geophysical Fluid Dynamics Laboratory (GFDL-ESM2M), National Oceanic and Atmospheric Administration (NOAA), which is one of the most robust models considering interactions between each sphere.

Land-use harmonization products provided by the IPCC Fifth Assessment Report give opportunities for estimating the impacts of a wide range of land-use trends on long-term terrestrial ecosystem processes[27]. The land-use harmonization dataset (LUH; http://luh.umd.edu/data.shtml) provides the annual land-use grid dataset from a long-term historical period and also provides the future land-use predictions under the different RCP scenarios (CMIP5). The fraction of each land-use type is described on a 0.25° grid in the LUH report, with the historical reconstruction period and four land-use change scenarios for future predictions. We choose the two representative concentration pathways, RCP4.5 and RCP8.5, which correspond to the NEX-GDDP climate data. LUH provides seven land-use types (primary forest, secondary forest, pasture, crop, primary non forest, secondary non forest, and urban) and we reclassified each LUH land-use report into five different broad land cover types (forest, grass, non forest, crop, and urban) in each pixel.

For the spatial distribution of global carbonate rock, we use the v3.0 version world map of carbonate rock outcrops provided by the Geography and Environmental Science Department, University of Auckland (http://:www.sges.aukland.ac.nz/sges_research/karst.shtm). This map only displays the outcrop of karstic solid rocks. It does not include carbonate rock types that are covered by later consolidated strata. The carbonate rock types in the natural environment consist chiefly of limestone ($CaCO_3$) and dolostone ($Ca(Mg)CO_3$). Due to the uncertainties of precisely distinguishing limestone from globally less common dolostone in the geological maps, we calculated CCSF by assuming that all carbonate outcrops are calcite in this study.

Atmospheric $CO_2$ ($CO_{2atm}$) is also an important factor in the air–water–rock system. We added $CO_{2atm}$ as an additional parameter for both historical and future emissions following the two pathways (RCP4.5 and RCP8.5). The historical $CO_{2atm}$ trends and different future emission prediction data (til 2100) were obtained from Potsdam Institute for climate impact research (http://www.pik-potsdam.de/~mmalte/rcps/index).

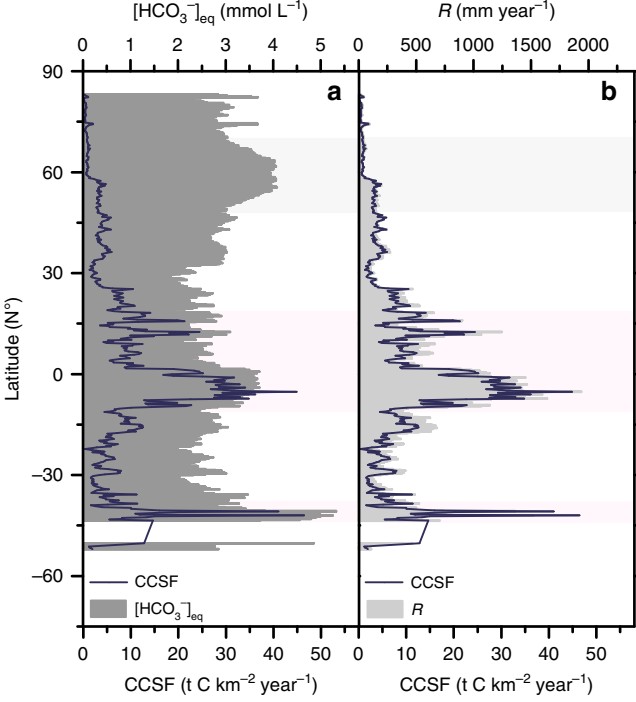

**Fig. 5 Latitudinal CCSF variation with relevant variables.** CCSF (carbonate weathering carbon-sink flux, blue line in **a** and **b**) in relation to **a** $[HCO_3^-]_{eq}$ (dark grey-shaded area) and **b** $R$ (runoff, light-gray shaded area) in the historical period (1950–2005). CCSF shows a significant positive relationship to runoff ($R$) across all latitudes.

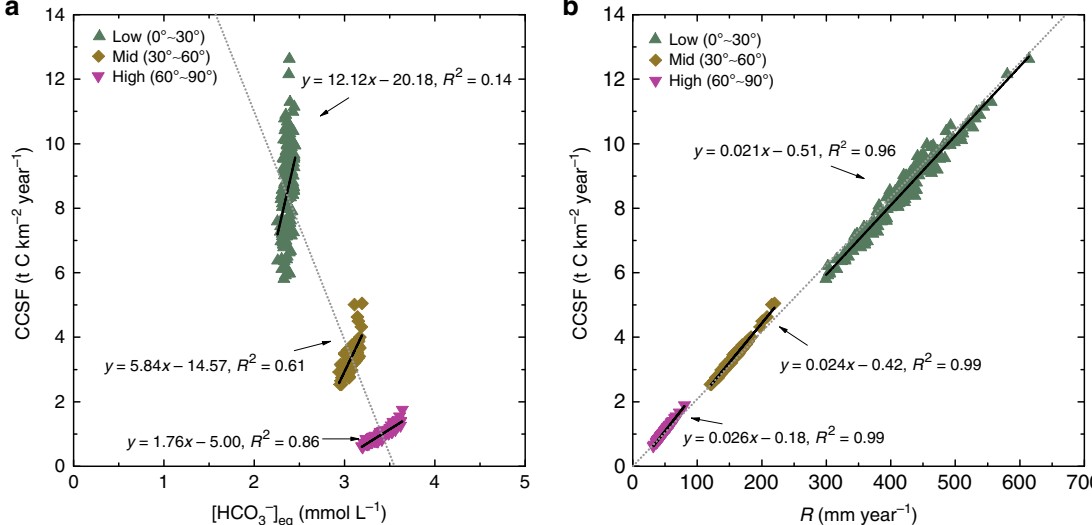

**Fig. 6 Relationship between CCSF and relevant variables. a** CCSF (carbonate weathering carbon-sink flux) and $[HCO_3^-]_{eq}$, and **b** CCSF and $R$ (runoff) in the historical period (1950–2005), normalized into three global latitudinal zones. Runoff dominates the CCSF variation across different latitudinal zones. $[HCO_3^-]_{eq}$ shows a high ($R^2 = 0.86$) positive relation with CCSF only in high latitudes (60°–90°), where runoff is low.

**Calculating equilibrium [Ca²⁺] in a karst system**. The calcium equilibrium concentration $[Ca^{2+}]_{eq}$ [mol m$^{-3}$] for a solution saturated with respect to calcite can be derived to very high accuracy from the analytical expression[9]:

$$[Ca^{2+}]_{eq}^3 = \frac{K_1 K_C K_H}{4 K_2 \gamma_{Ca^{2+}} \gamma_{HCO_3^-}^2} pCO_2 \qquad (2)$$

where $K_1$, $K_2$, $K_c$, and $K_H$ are the temperature-dependent equilibrium constants for the chemical reactions, $\gamma_{Ca2+}$ and $\gamma_{HCO3^-}$ are the activity coefficients for calcium and bicarbonate, respectively, and $pCO_2$ (in atm) is the carbondioxide partial pressure[9].

**Calculation of pCO₂ for carbonate weathering**. $CO_2$ is a key driving factor for carbonate dissolution. It is present in the atmosphere and will be enhanced by soil respiration. The $pCO_2$ along the soil–rock or atmosphere–rock interface controls the saturation state of carbonate chemistry for ground water, thereby determining the amount of carbonate that can be dissolved in a karst aquifer[6,9]. In this study, soil $pCO_2$ is calculated by the method given by the Gwiazada and Broecker[28] and more recently modified by Gaillardet et al.[6], who conclude that $CO_2$ production by respiration in the root zone ($CO_2^{pr}$ in g C m$^2$ yr$^{-1}$) can be assumed to be 75% of the ecosystem net primary production (NPP). Meanwhile, a power function is used to define the $pCO_2$ profile by solving the complete $CO_2$ diffusion equation in soil. It is assumed at the basis of the root production zone, the soil $CO_2$ reaches maximum and becomes constant below this horizon: soil $pCO_2$ can thus be expressed as a function of atmospheric $CO_2$ concentration, temperature, and NPP, shown as below[6]:

$$pCO_{2(Soil)} = pCO_{2(atm)} + \frac{A \times 0.75 \times NPP}{T^2} \qquad (3)$$

where $A = 1.03 \times 10^6$, a conversion unit constant, $pCO_{2atm}$ is the atmospheric $CO_2$ pressure in ppmv. NPP is net primary productivity in grams of dry matter per meter square per year (g m$^{-2}$ yr$^{-1}$), $T$ is the surface temperature expressed in K and $pCO_2$ (soil) is the maximum $CO_2$ pressure reached below the root zone in ppmv. In the original version of Eq. (3), the soil $pCO_2$ also depends on the mean root depth, soil porosity, and tortuosity[13,28]. However, the rooting depths of global vegetations vary in different plants species. According to the data from 475 soil profiles around the world, the majority of rooting depths among different vegetation types are similar in most regions[29]. As suggested by Schenk and Jackson[29], for predictions on a global scale, it may be undesirable to assign fixed rooting depths to different vegetation types. Moreover, it has been found that the forest, scrub, and grass have a similar soil horizon 50–80 cm that contain 90% of the root biomass[30], and this horizon is matching well with the mean root depth in the soil $pCO_2$ model, we mentioned above[28]. On the other hand, recent studies show that different land use may have similar soil porosity (forest, cultivated land, and grassland), even after revegetation or deforestation for agriculture activities[31,32]. Human activities may impact the soil porosity at the surface soil layers[31], but may not alter the subsurface soil layers where the soil $pCO_2$ reaching maximum. Soil tortuosity depends on porosity as found by Jin and Jury[33]. Due to these evidences, we think the recommended soil porosity and tortuosity by Goddéris et al.[13] and Gaillardet et al.[6] are feasible in present research. Here, by using Eq. (3) we estimate the soil $pCO_2$ differences by NPP. We used the Miami model to calculate NPP in forest ecosystems[34]. The model assumes that the climate limits the vegetation primary production and that NPP increases with both increasing temperature and increasing precipitation:

$$NPP_{(T,P)} = min \left\{ \begin{array}{c} \frac{3000}{1 + e^{1.315 - 0.119 \times T}} \\ 3000 \times (e^{-0.000664 \times P}) \end{array} \right\} \qquad (4)$$

where NPP is net primary production for ecosystem (NPP is the amount of organic matter in g of dry matter m$^{-2}$), $T$ [C] is the annual mean surface temperature, and $P$ [m s$^{-1}$] is the annual mean precipitation. Though recent studies found that the response of NPP to changes in precipitation and temperature varies between ecosystems, the Miami model results are used to obtain a close approximation of NPP values in forest ecosystems, and probably overestimates NPP in non-tree-dominated (grass, shrub, and crop) ecosystems that are largely controlled by precipitation variation and soil water content. In order to quantify the soil $pCO_2$ differences between different land use/land cover, we employed another model which includes the NPP estimation in ecosystems without trees. This NCEAS model[35] is as follows:

$$NPP_{(non-tree)} = 6116 \times [1 - \exp(-6.05 \times 10^{-5} \times P)] \qquad (5)$$

where NPP (non-tree) is net primary production (g C m$^2$ yr$^{-1}$) in non-tree-dominated ecosystems, and $P$ the annual mean precipitation. Due to land-use reclassification in the LUH dataset, we used the NCEAS model to calculate the NPP for grass, non forest, and crop.

The $pCO_2$ in urban areas is set equal to the atmospheric level due to the absence of soil. In addition, the $pCO_{2atm}$ trends under different RCP scenarios (1950–2100) given by the Potsdam Institute for climate impact research http://www.pik-potsdam.de/~mmalte/rcps/ are integrated into our model. It is assumed to be 320 ppm in 1950 AD according to the existing records. In the 2100, as predicted by RCP4.5 and RCP8.5, $pCO_{2atm}$ will reach 538 ppm and 945 ppm, respectively.

**Runoff variation estimation by climate and land-use change**. In our prior CCSF modeling studies, long-term runoff changes were calculated by resolving the

balance equation between precipitation and evapotranspiration[12]. During the past few decades, the historical global water cycle seems to have strengthened[36]. For a long time, water yield has been mainly considered to depend on natural factors but recent studies have emphasized that anthropogenic factors, such as land-use and land-cover changes can be another factor driving runoff perturbations in areas with major human interventions[21]. Generally, in forest-dominated catchments, the evapotranspiration is higher than in grass-dominated catchments under similar climatic conditions, because of the different water consumption capacities of plants[37]. Agricultural activities and urbanization can also alter the vegetation cover, soil properties and thereby change water yield and runoff patterns. Here, we use the model recommended by Zhang et al.[37] to separately estimate the evapotranspiration of forest and of grass lands. In addition, in order to estimate the hydrological changes of other land-cover/land-use types in the LHU dataset, we additionally introduce three extended models (for crop, non forest, and urban) that are based on the standard function given by Zhang et al.[37]. Our extension functions are based on a three-year water balance study in a karst simulation test site that detects the water yields of five different land uses in karst terrain[17]. The final modified model can be expressed as:

$$ET_{sum} = fET_f + gET_g + nET_n + cET_c + uET_u \qquad (6)$$

where $ET_{sum}$ (mm) is the total annual evapotranspiration, $f$, $g$, $n$, $c$, and $u$ are the ratios of forest, grass, non forest, crop, and urban cover in each pixel, respectively ($f+g+n+c+u=1$), and $ET_f$, $ET_g$, $ET_n$, $ETc$, and $ET_u$ (mm) are the corresponding annual evapotranspiration from different land uses.

**Maximal potential dissolution method used for CCSF estimate**. To obtain the global CCSF variations for long-time periods, we replace $R$ in Eq. (1) by $R = P$-ET, with $P$ the total precipitation (m yr$^{-1}$) and ET the evapotranspiration (m yr$^{-1}$). We assume that the total dissolved carbon can be approximated by the bicarbonate alone, $[DIC] = [HCO_3^-]$, which is valid for pH values around 8. In this pH range, reduced electroneutrality states that for each bicarbonate molecule, two calcium atoms are present, thus $[HCO_3^-] = 2[Ca^{2+}]$. Thus, after multiplying Eq. (1) with the atomic mass of carbon, 12 g mol$^{-1}$, Eq. (1) can be reformulated to the annual CCSF (t C km$^{-2}$ yr$^{-1}$; refs. [11,12]):

$$CCSF = 12(P - ET)[Ca^{2+}]_{eq} \qquad (7)$$

where $[Ca^{2+}]_{eq}$ is the concentration of calcium ion at equilibrium (mol m$^{-3}$). The annual TCS (t C yr$^{-1}$) for a given karst area can be calculated by:

$$TCS = 12(P - ET)S[Ca^{2+}]_{eq} \qquad (8)$$

where $S$ [km$^2$] is the land surface area of the carbonate outcrops.

## Data availability

The authors declare that the data supporting the findings of this study are publicly available in the web pages provided in the article. The equilibrium model and all relevant data are available from the corresponding author upon request. The source data underlying Figs. 1, 3–6 are provided as a Source Data file.

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

## Acknowledgements

We thank Prof. Dr. Derek Ford for insightful comments on previous versions of this manuscript. This study has been financially supported by the National Natural Science Foundation of China (41430753, U1612441, and 41921004) and the Strategic Priority Research Program of Chinese Academy of Sciences.

## Author contributions

Z.L. was the leader of the project financially supported by NSFC. G.K. and Z.L. designed the modelling. S.Z. ran the model and wrote the paper.

## Competing interests

The authors declare no competing interests.
