## [Peer Review File · Nature Communications]

Reviewers' comments:

Reviewer #1 (Remarks to the Author):

This contribution explores the response of the global weathering flux of carbonate outcrops to global climate and land use changes. The method is straightforward: it is assumed that the water flowing on carbonate outcrops is at chemical equilibrium with respect to calcite (certainly a valid hypothesis). Second, they superimpose runoff, temperature, and land use maps to estimate the amount of HCO_3^- released by carbonate weathering to the ocean, and corresponding to the atmospheric CO_2 consumption. To do so, the model uses several parametric laws.

The main finding of this study is, I think, the importance of the impact of land use changes on the weathering processes. There is another conclusion where the authors suggest that carbonate weathering may become a negative feedback, potentially tempering the accumulation of anthropogenic CO_2 into the atmosphere (at least at short timescales, below the mixing time of the ocean).

I think the paper is well written, and easy to follow. It deserves publication, but I have several questions, the most critical being the third one

1. Regarding the role played by carbonate weathering in the global carbon budget, the author should mention somewhere that carbonate weathering accounts for about 0.2GtC/yr (Gaillardet et al., 1999). Although certainly not negligible, this flux is probably a bit low to exert a significant negative feedback on the accumulation of CO_2 in the atmosphere by anthropic activities.

2. The question of the role of other acids than carbonate acid is mentioned at two places in the text. Once in the legend of table 1, and then when discussing the impact of agriculture of carbonate weathering around line 300. As correctly stated by the authors, when carbonate rocks are attacked by sulfuric acids, the overall budget (at centennial to millennial scale) is a net degassing of carbon into the atmosphere. I know this is not the focus of this contribution, but can you give any estimate of this flux in the future? I'm not sure it is feasible, so this question might be premature. There are some published works for the mountainous area (check Torres et al. 2016 in EPSL), but the agricultural contribution is probably much bigger. Answering this point is certainly not mandatory.

3. The question of the below ground CO_2 level is more complex. Finally, all works calculating below ground CO_2 at the global scale are relying on the equation proposed by Gwiazda and Broecker 1994. In fact, the equation gives the maximum CO_2 level at the basis of the CO_2 production zone (Van Bavel 1951, cited in Gwiazda and Broecker). It is the equation 2 in G&B, combined with equation 3 in G&B. The equation 3 in the present paper groups all the parameters, except the temperature and the CO_2 production, within the parameter A, which is interpreted as a conversion unit constant. But it is not the case. Back to Van Bavel (1951) and explained in Godderis et al 2010 GCA, the present A factor includes the thickness of the CO_2 production layer (probably equal to the thickness of the root zone), and the porosity and tortuosity of the soil. Can you take those factors into consideration? They are not constant at the global scale. I think these are critical for land use purposes (thickness of the root zone and physical properties of the soil are heavily impacted when moving from a forest to field).

This paper is worth publishing in Nature Comm, since it deals with processes that matters for many geoscientists. But point 3 should be solved or discussed first.

Reviewer #2 (Remarks to the Author):

Zeng et al. present a compelling and straight forward modeling framework for demonstrating the

sensitivity of carbonate weathering to potential changes in future climatic conditions. Their main goal is to underscore that this a substantial part of the terrestrial carbon cycle that should be incorporated into global climate models. Their simple compilation demonstrates that carbonate weathering will increase ~9-17% (depending on RCP selected). Overall this manuscript is impactful and will be of interest to the readers of nature communication and is written in a manner that is accessible to broad audiences. Given the nature of such simplified arguments, the manuscript would be stronger if all assumptions were clearly identified and explored. Below I list several comments that should be addressed before publication.

Land use change can prompt changes in subsurface flow paths and mineral water interaction, thus the resultant fluxes of solutes from landscapes. Given thermodynamic-controls on carbonate weathering, and the findings that water fluxes through the landscape will have the biggest control on $[\text{HCO}_3^-]_{\text{eq}}$, it would be good to at least mention that this not accounted for in land use change dynamics.

Given temperature and soil respiration are both controlling carbonate weathering it would be good to have maps of differences in soil CO₂ between "present" and "future" scenarios to see the degree to which each portion of the system is changing, given temperature alone does not have much effect on CCFS. It could be as simple as adding a curve of projected soil CO₂ with latitude for future and past conditions along with the precipitation and temperature one shown in figure 4.

What is the net effect of this increase weathering on global biogeochemical cycles, what does it mean for ocean systems? What potential impact does this have for atmospheric CO₂ concentrations? Can you put this back in the context of the global C budget at the end of the manuscript?

Minor comments:

Page 2 Line 6: Add reference to the 0.5 value

Page 2 Line 14: I think the implications of DIC flux as a result of nitric acid weathering due to fertilizers should be mentioned. It is mentioned at the end of the discussion on page 13 but the lack of inclusion should be mentioned earlier.

Page 2 Line 15: The major point made in the Konza Prairie study is that groundwater CO₂ concentration are increasing at rate much higher than atmospheric CO₂. A 2019 paper suggests the possible reasons for this increase, this statement should be correct to just implicate that groundwater storage of CO₂ in this system is increasing.

Macpherson, G. L., Sullivan, P. L., Stotler, R. L., & Norwood, B. S. (2019). Increasing groundwater CO₂ in a mid-continent tallgrass prairie: Controlling factors. In E3S Web of Conferences (Vol. 98, p. 06008). EDP Sciences.

Reviewers' comments:

Reviewer #1 (Remarks to the Author):

This contribution explores the response of the global weathering flux of carbonate outcrops to global climate and land use changes. The method is straightforward: it is assumed that the water flowing on carbonate outcrops is at chemical equilibrium with respect to calcite (certainly a valid hypothesis). Second, they superimpose runoff, temperature, and land use maps to estimate the amount of HCO_3 released by carbonate weathering to the ocean, and corresponding to the atmospheric CO_2 consumption. To do so, the model uses several parametric laws.

The main finding of this study is, I think, the importance of the impact of land use changes on the weathering processes. There is another conclusion where the authors suggest that **carbonate weathering may become a negative feedback, potentially tempering the accumulation of anthropogenic CO_2 into the atmosphere (at least at short timescales, below the mixing time of the ocean).**

I think the paper is well written, and easy to follow. It deserves publication, but I have several questions, the most critical being the third one.

1. Regarding the role played by carbonate weathering in the global carbon budget, the author should mention somewhere that carbonate weathering accounts for about 0.2GtC/yr (Gaillardet et al., 1999). Although certainly not negligible, this flux is probably a bit low to exert a significant negative feedback on the accumulation of CO_2 in the atmosphere by anthropic activities.

Re: thanks, the reference has been cited, see lines 31-33 on page 1 in the revised MS with marks. It shows that even if the value 0.2 was used, it accounted for approximately 7% of the estimated terrestrial carbon sink, which is not negligible.

2. The question of the role of other acids than carbonate acid is mentioned at two places in the text. Once in the legend of table 1, and then when discussing the impact of agriculture of carbonate weathering around line 300. As correctly stated by the authors, when carbonate rocks are attacked by sulfuric acids, the overall budget (at centennial to millennial scale) is a net degassing of carbon into the atmosphere. I know this is not the focus of this contribution, but can you give any estimate of this flux in the future ? I'm not sure it is feasible, so this question might be premature. There are some published works for the mountainous area (check Torres et al. 2016 in EPSL), but the agricultural contribution is probably much bigger. Answering this point is certainly not mandatory.

Re: the role of other acids than carbonate acid has been estimated by Lerman et al. (2007), who found the higher input rates of H_2SO_4 may increase the dissolved ionic concentrations in river waters by about 13%, without significantly affecting the CO_2 consumption in weathering, and Perrin et al. (2008), who suggested this CO_2 source by nitric acid due to agriculture contribution is not negligible since it could reach 6–15% of CO_2 uptake by natural silicate weathering and could consequently partly counterbalance this natural CO_2 sink. Sorry, to give an estimate of this flux in the future may be difficult, which, as you know, is out of the focus of this contribution. See lines 1-9 on page 18 in the revised MS with marks.

References

- Lerman, A., Wu, L.L., Mackenzie, F.T., 2007. CO_2 and H_2SO_4 consumption in weathering and material transport to the ocean, and their role in the global carbon balance. *Mar. Chem.* 106, 326–350.
- Perrin, A., Probst, A., Probst, J.L., 2008. Impact of nitrogenous fertilizers on carbonate dissolution in small agricultural catchments: implications for weathering CO_2 uptake at regional and global scales. *Geochim. Cosmochim. Acta* 72, 3105–3123.

3. The question of the below ground CO_2 level is more complex. Finally, all works calculating below ground CO_2 at the global scale are relying on the equation proposed by Gwiazda and Broecker 1994. In fact, the equation gives the maximum CO_2 level at the basis of the CO_2 production zone (Van Bavel 1951, cited in Gwiazda and Broecker). It is the equation 2 in G&B, combined with equation 3 in G&B. The equation 3 in the present paper groups all the parameters, except the temperature and the CO_2 production, within the parameter A, which is interpreted as a conversion unit constant. But it is not the case. Back to Van Bavel (1951) and explained in Godderis et al 2010 GCA, the present A factor includes the thickness of the CO_2 production layer (probably equal to the thickness of the root zone), and the porosity and tortuosity of the soil. Can you take those factors into consideration? They are not constant at the global scale. I think these are critical for land use purposes (thickness of the root zone and physical properties of the soil are heavily impacted when moving from a forest to field).

Re: yes, you're right. But we used the soil pCO_2 equation given by Gallerdet (2018) who considered the recommended values of soil porosity and tortuosity from Godderis (2010) and mean root zone depth from Gwiazada & Broecker(1994). Gallerdet (2018) used a factor A to reduce original equation and to calculate the global soil pCO_2 and equilibrium Ca^{2+} concentration. The reasons why we used this version in our research are as following:

(1) Although major **rooting depths** vary among vegetation species, it is still questioned to give different values for each land cover type, such as forest, grass, corps or shrub. Some tree species probably have much deeper roots

than grass or shrub. However, according to the data from 475 soil profiles around the world, the 95% of rooting depths among different vegetation types were similar in most regions (Schenk and Jackson, 2002). As suggested by Schenk and Jackson (2002), for predictions on a global scale, it may be undesirable to assign fixed rooting depths to different vegetation. Moreover, it has been found that forest, scrub, grass have a similar soil horizon 50-80cm that contain 90% of the root biomass (Schulze et al, 1996), and this level was matching well with the mean root depth setting in the model we used. Therefore, we believe the recommended mean root depth is feasible for global soil pCO₂ modelling in our study.

(2) A growing evidence showed that different land use has similar **soil porosity** (forest, farmland and grassland), even after revegetation or deforestation by agriculture activities (Jiao et al, 2010; Khaledian et al, 2016; Evrendilek et al, 2004; Wang et al, 2014). Human activities may impact the soil porosity at the surface soil layers. However, it may not alter the subsurface soil layers (Jiao et al, 2010) where the soil pCO₂ reach maximum. In addition, a global spatial database offering the global deep soil porosity seems unavailable. Therefore, it was feasible to follow the original setting in Gallardet (2018).

(3) Soil tortuosity depends on porosity as found by Yin (1996). Due to the similar reason to porosity, we also adopted the recommended value given by Gwiazda & Broecker(1994) and Godderis (2010).

We accepted this nice suggestion and added more details and discussions in method introduction, see lines 3-15 on page 20 in the revised MS with marks.

References

- Evrendilek, F., Celik, I. & Kilic, S. Changes in soil organic carbon and other physical soil properties along adjacent Mediterranean forest, grassland, and croplands in Turkey. *J Arid Environ* **59**:743–752 (2004).
- Gaillardet, J., Calmels, D., Romero-Mujalli, G. Z. & Hartmann, J. Global climate control on carbonate weathering intensity. *Chem. Geol.* [10.1016/j.chemgeo.2018.05.009](https://doi.org/10.1016/j.chemgeo.2018.05.009) (2018).
- Godd ris, Y., Williams, J. Z., Schott, J., Pollard, D. & Brantley, S. L. Time evolution of the mineralogical composition of Mississippi Valley loess over the last 10kyr: Climate and geochemical modelling. *Geochim Cosmochim Acta* **74**, 6357-6374 (2010).
- Gwiazda, R. H. & Broecker, W. S. The separate and combined effects of temperature, soil pCO₂ and organic acidity on silicate weathering in the soil environment: Formulation of a model and results. *Glob. Biogeochem. Cycles.* **8**, 141-155 (1994).
- Jiao, F., Wen, Z.M. & An, S.S. Changes in soil properties across a chronosequence of vegetation restoration on Loess Plateau of China. *Catena*, **86** (2), 100-166 (2011).

Khaledian, Y., Kiana, F., Ebrahimi, S., Brevik, E & Aitkenhead-Peterson, J. Assessment and monitoring of soil degradation during land use change using multivariate analysis. *Land Degrad Dev.* **28**, 128-141 (2016).

Schenk, H. & J. Jackson, R.B. The global biogeography of roots. *Ecol. Monogr.* **72**, 311-328 (2002).

Schulze, E.-D., Mooney, H.A., Sala, O. E., Jobbagy, E. & Buchmann, N. Rooting depth, water availability, and vegetation cover along an aridity gradient in Patagonia. *Oecologia.* **108**, 503-511 (1996).

Wang, B., Zhang, G.H., Shi, Y.Y. & Zhang, X.C. Soil detachment by overland flow under different vegetation restoration models in the Loess Plateau of China. *Catena* **116**, 51–59 (2014).

Jin, Y. & Jury, W.A. Characterizing the dependence of gas diffusion coefficient on soil properties. *Soil Sci. Soc. Am. J.* **60**, 66-71 (1996).

This paper is worth publishing in Nature Comm, since it deals with processes that matters for many geoscientists. But point 3 should be solved or discussed first.

Re: as shown above.

Reviewer #2 (Remarks to the Author):

Zeng et al. present **a compelling and straight forward modeling framework for demonstrating the sensitivity of carbonate weathering to potential changes in future climatic conditions.** Their main goal is to underscore that this a substantial part of the terrestrial carbon cycle that should be incorporated into global climate models. Their simple compilation demonstrates that carbonate weathering will increase ~9-17% (depending on RCP selected). Overall **this manuscript is impactful and will be of interest to the readers of nature communication and is written in a manner that is accessible to broad audiences.** Given the nature of such simplified arguments, the manuscript would be stronger if all assumptions were clearly identified and explored. Below I list several comments that should be addressed before publication.

Land use change can prompt changes in subsurface flow paths and mineral water interaction, thus the resultant fluxes of solutes from landscapes. Given thermodynamic-controls on carbonate weathering, and the findings that water fluxes through the landscape will have the biggest control on [HCO₃]⁻eq, it would be good to at least mention that this not accounted for in land use change dynamics.

Re: Yes, we have mentioned it, see the line 41-44 on page 17 in the revised MS with marks.

Given temperature and soil respiration are both controlling carbonate weathering it would be good to have maps of differences in soil CO₂ between “present” and “future” scenarios to see the degree to which each portion of the system is changing, given temperature alone does not have much effect on CCFS. It could be as simple as adding a curve of projected soil CO₂ with latitude for future and past conditions along with the precipitation and temperature one shown in figure 4.

Re: it is a nice suggestion! We added a new graph Figure 3(a) to explain the differences of soil pCO₂ in historical period and future scenarios. In addition, we also plotted the latitudinal soil pCO₂ distribution, comparing with [HCO₃⁻]_{eq}, as shown in Figure 4(d). We believe such figures can better describe the changes in soil pCO₂ dynamics as reviewer suggested. The text to explain and describe these changes can be found in the revised MS with marks, see the line 4-9 on page 6 in the revised MS with marks.

What is the net effect of this increase weathering on global biogeochemical cycles, what does it mean for ocean systems? What potential impact does this have for atmospheric CO₂ concentrations? Can you put this back in the context of the global C budget at the end of the manuscript?

Re: The increased carbonate weathering flux from terrestrial waters to oceans may promote the biological carbon pump in ocean systems, leading to net carbon sequestration in the ocean. It can probably be a considerable carbon sink against the rising atmospheric CO₂ in future. According to this suggestion, we put these potential impacts at the end of the manuscript, see lines 32-35 on page 17 in the revised MS with marks.

Minor comments:

Page 2 Line 6: Add reference to the 0.5 value

Re: A reference has been added, see line 4 on page 2 in the revised MS with marks please.

Page 2 Line 14: I think the implications of DIC flux as a result of nitric acid weathering due to fertilizers should be mentioned. It is mentioned at the end of the discussion on page 13 but the lack of inclusion should be mentioned earlier.

Re: We added some text here to mention the impacts of nitric acid at the introduction part, see lines 15-16 on page 2 in the revised MS with marks.

Page 2 Line 15: The major point made in the Konza Prairie study is that groundwater CO₂ concentration are increasing at rate much higher than atmospheric CO₂. A 2019 paper suggests the possible reasons for this

increase, this statement should be correct to just implicate that groundwater storage of CO₂ in this system is increasing.

Macpherson, G. L., Sullivan, P. L., Stotler, R. L., & Norwood, B. S. (2019). Increasing groundwater CO₂ in a mid-continent tallgrass prairie: Controlling factors. In E3S Web of Conferences (Vol. 98, p. 06008). EDP Sciences.

Re: thanks, we updated the reference and corrected the findings in Konza Prairie, see lines16-20 on page 2 in the revised MS with marks.

REVIEWERS' COMMENTS:

Reviewer #1 (Remarks to the Author):

I think the authors have answered the questions I raised in the first round of review. I also thank the authors for discussing further the assumptions related to the calculation of the below ground CO₂ level at the global scale. I have no other questions and suggest that this revised contribution should be published in Nature Comm.

Best regards

Reviewer #2 (Remarks to the Author):

You done a nice job of responding to all of the comments and produced a very nice manuscript. I think this will be of interest to nature communications.

November 25, 2019

Response to referees

There are no further concerns from the two reviewers as shown below:

REVIEWERS' COMMENTS:

Reviewer #1 (Remarks to the Author):

I think the authors have answered the questions I raised in the first round of review. I also thank the authors for discussing further the assumptions related to the calculation of the below ground CO₂ level at the global scale. I have no other questions and suggest that this revised contribution should be published in Nature Comm.

Best regards

Reviewer #2 (Remarks to the Author):

You done a nice job of responding to all of the comments and produced a very nice manuscript. I think this will be of interest to nature communications.

We thank the time and expertise from the reviewers.

Zaihua Liu (Prof. Dr.)